# Characteristics of Calcium Isotopes at Different Water Depths and Their Palaeoenvironmental Significance for Carbonate Rocks of the Permian-Triassic Boundary in Chibi, Southern China

**Hongyu Zhao [1,2] and Junhua Huang [2,\*]**

1 School of Earth Sciences, China University of Geosciences, Wuhan 430074, China
2 State Key Laboratory of Geological Processes and Mineral Resources, China University of Geosciences, Wuhan 430078, China
\* Correspondence: jhhuang@cug.edu.cn

**Abstract:** Calcium isotopes of carbonate rocks can trace calcium cycles and record changes in the marine environment. As published calcium isotope profiles of carbonate rocks at the Permian-Triassic boundary are rare, comparative studies on deep-water profiles were lacking for the major extinction event that occurred during this time. We present sections of different water depths in the Chibi area of southern China that we have selected for a comparative study. We analyzed carbon isotopes, calcium isotopes, as well as major and trace elements of carbonates from two sections (Chibi North and Chibi West) to obtain information on the volcanic activity, ocean acidification, as well as sea level rise and fall in the Chibi area during the mass extinction period. All carbon and calcium isotopes of carbonates from both sections are all negative after the mass extinction boundary. Carbonates from the Chibi North section have higher $\delta^{44/40}Ca$ values and lower Sr/Ca ratios than the rocks from the Chibi West section. We propose that the negative bias of the calcium isotopes in the two sections result from diagenesis. Diagenesis transforms primary aragonite into calcite, showing the characteristics of high $\delta^{44/40}Ca$ value and low Sr/Ca. By comparing our data with three published profiles of shallow-water carbonate rock, we recognize that calcium isotopes record gradients at different water depths. In the slope environment, the enhancement of pore fluid action near the coast caused an increase of the fluid buffer alteration, and we propose that a regression event occurred in the Chibi region during the Late Permian.

**Keywords:** Permian-Triassic boundary; South China; calcium isotope; diagenesis; regression

## 1. Introduction

The Permian-Triassic extinction event was the most severe mass extinction event in the history of the Earth [1]. It was characterized by widespread ocean hypoxia [2,3], ocean acidification [4,5], intense volcanic activity [6–8] rapid temperature rise [9,10], and sea level rise and fall [11,12]. Carbonate minerals are the most important calcium sinks in the oceans, with high calcium content and wide distribution [13]. Therefore, calcium isotopes of carbonate rocks are ideally suited to characterize the nature of the global changes during the Permian-Triassic extinction event [14]. Rapid short-term fluctuations in seawater calcium isotope values during the geological history respond to changes in the climatic environment [15]. The calcium isotopes of carbonate rocks from the Permian-Triassic period record a global negative calcium isotope shift, which may be caused by ocean acidification [5]. However, the negative calcium isotope shift of carbonate rocks from different depositional environments exhibits some variability, which may be related to diagenesis in different environments.

The chemical and isotopic composition of carbonate sediments can be partially or completely reset by diagenesis during the deposition of the carbonate rocks [16], with early

marine diagenesis and syn-deposition occurring prior to lithification. The recrystallization process results in the growth of new carbonate crystals and the conversion of aragonite to calcite [17]. The calcite fractionation factor is a function of the sedimentation rate, and in slowly recrystallizing marine sediments, the fractionation factor is $\pm 0$. Thus, newly grown calcite has a higher value than unaltered calcite due to the addition of Ca from the seawater. This effect is particularly prevalent in deep-sea sediments and in carbonate-rich sediments of platform environments [18–20]. Aragonite tends to have much lower $\delta^{44/40}$Ca values than calcite, and thus the change in the carbonate sediment mineralogy from aragonite to calcite causes an increase of the $\delta^{44/40}$ Ca values [3,20].

Carbonate rocks are subjected to fluid buffer alteration caused by fluid flow [16]. Pore fluid circulation is strongest at the basin margins near the landward direction and slowest in the basin centers near the sea [21]. Therefore, we propose that the carbonate sedimentation at different water depths at the Permian-Triassic boundary will affect the calcium isotope values of the carbonate rocks because of different pore-fluid circulation. Previous studies on the Permian-Triassic boundary have focused on calcium isotope studies on rocks from shallow-water carbonate terraces [5,14,22]. Profiles for the transition from shallow to deep water were lacking so far. Therefore, we have selected two representative carbonate sedimentation sections at different water depths in the same area, termed Chibi North and Chibi West sections, to analyze their carbon, oxygen, and calcium isotopes as well as major and trace elements, to better understand the significant environmental changes that occurred at the Permian-Triassic boundary from different water depths.

## 2. Geological Setting

According to the stratigraphic zoning, the region of Chibi in southern China belongs to the Lower Yangtze stratigraphic subdivision of the Yangtze stratigraphic region. In the Aurignacian to Middle Permian, marine sedimentation prevailed and mainly carbonate rocks interspersed with clastic rocks were deposited. During the Late Permian, the present-day Chibi area was situated on the slope of the northern margin of the Central Yangtze Plateau (Figure 1).

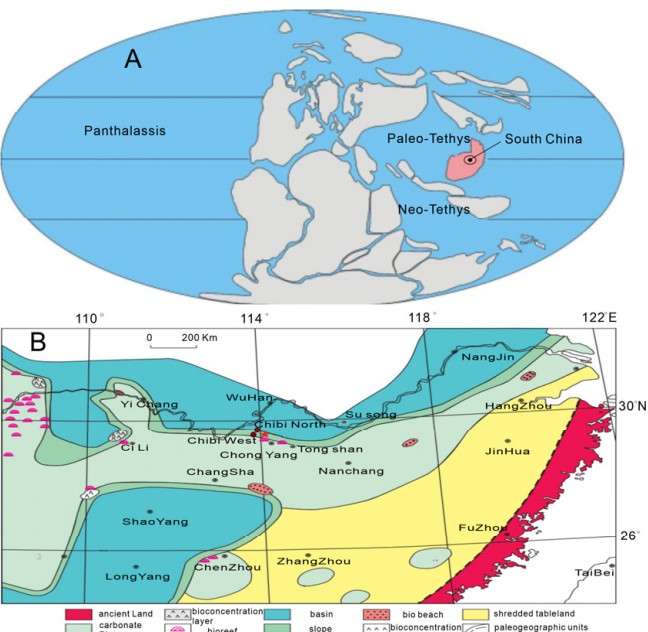

**Figure 1.** (**A**) Global paleogeography at the Permian-Triassic boundary, showing the location of South China (modified from [23]) (**B**) Paleogeographic context of the Late Permian Changhsingian period in South China, with the location of Chibi (modified from [24]).

The region to the north of the Chibi area is dominated by the the Late Permian Dalong Formation, a typical deep-water siliciclastic basin [25]. The area from Chongyang, Hubei to Tong Shan south of Chibi formed in a typical shallow water carbonate terrace environment. Blocky bioclastic tuffs were mainly deposited and locally sponge reefs are developed [26]. The region to the west of Chibi represents a paleogeographic transition from the edge of the shallow marine terrace to the deep-water siliciclastic basin [27]. Chibi is located on the northern edge of the South China Plate, which was a small plate between the Pan-ocean and the Tethys Ocean during the Early Triassic. From the Late Palaeozoic to the Early Triassic, this branch trough continuously deposits a complete sequence of marine strata, therefore, it clearly records the evolution history of marine environment, biology and geotectonics at the turn of the ancient and Mesozoic [28]. Due to global geological events at the end of the Permian, marine life and the environment have changed dramatically [29,30]. On the deep water basin profile, some volcanic clay deposits can be seen in the Permian and Triassic strata [7,31], however, research suggests that these volcanic activities do not come from this region, but from the southwest and Southeast Asia of the South China plate [32]. The shallow water platform at the southern edge of the trough was affected by the drop of the sea [33,34] at the end of the Permian, resulting in a small loss of strata at the intersection of the Permian and Triassic systems, but the attitude of the upper and lower strata is consistent [12], indicating that no strong tectonic movement occurred in the area during this period, so sea level decline may be due to global climate or plate movements.The unique paleolocation of Chibi is conducive to study the impact of the diagenesis and the environmental context of carbonate rocks in shallow-water terranes and in deep-water siliciclastic basins at the Permian-Triassic boundary.

The Chibi North section (29°75′15″ N, 113°95′19″ E) is situated in the deep-water siliciclastic basin [23]. During the Lopingian period, the study area was located in the eastern tropics of the Paleotethys. A massive carbonate platform (Yangtze platform) was developed in the South China Block at this time. Adjacent to a deeper basin, siliceous carbonate and chert deposits were deposited further north during the Wuchiapingian. The Chibi North profile of the Wuchiapingian period comprises a sequence of shallow-water carbonate platform sediments. Due to the sea level rise, the depositional environment changed to a deeper slope environment in the latest Changhsingian and the successively sedimented strata include the Wuchiaping and Dalong formations, In the Early Triassic the Daye Formation was deposited.Gastropods in the Late Permian and Early Triassic strata are abundant and have strong biological disturbances.Outcrop pictures of Chibi North and Chibi West sections are shown in Figure 2.

The Chibi West section is located in Chibi City, Hubei Province, about 4 km west of the county (29°41′57″ N, 113°51′33″ E) [25]. Paleogeographically the Chibi West section comprises the transition zone from a shallow marine terrace to a deep-water basin. During the Late Permian to Early Triassic, the region was situated at the northern edge of the South China Plate and was part of the Yangtze carbonate plateau. The area comprises a gently sloping zone. The Late Permian to Early Triassic stratigraphy of the Chibi West section is well exposed, and the Permian-Triassic intersection is well recorded as a major biological extinction boundary. A thin grayish-black sequence of intermediate thickness of the Late Permian Dalong Formation with layered siliciclastic or siliciclastic-striped tuffs is exposed below the mass extinction boundary but above the line that marks the mass extinction boundary as the Early Triassic Daye Formation; a sequence of grey-black-black mudstone intersperses with marl at the bottom of the Daye Formation.

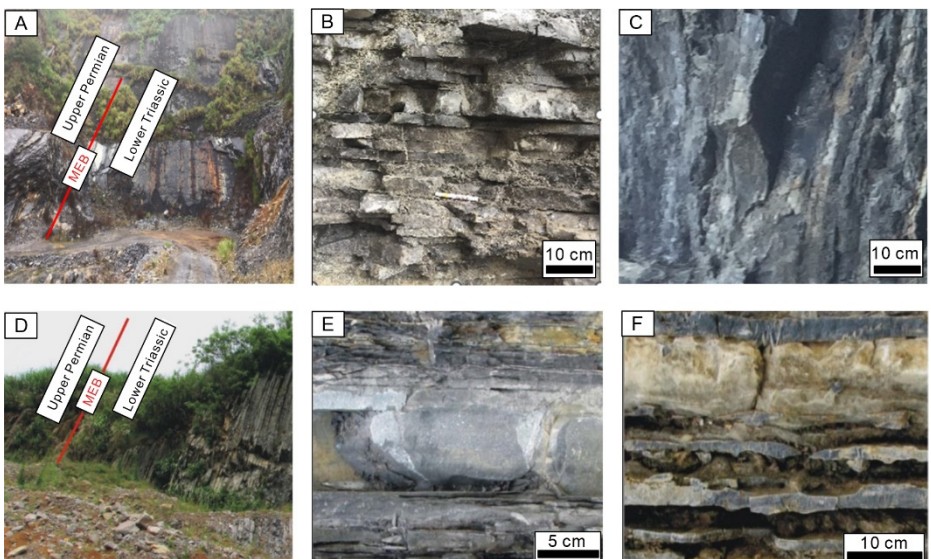

**Figure 2.** outcrop pictures of Chibi North and Chibi West sections. (**A**) outcrop picture of Chibi North section, (**B**) argillaceous limestone at the bottom of the Early Triassic Daye Formation, Chibi North section, (**C**) siliceous limestone in the top of the Late Permian Dalong Formation, Chibi North secion, (**D**) outcrop picture of Chibi West section, (**E**) black argillaceous limestone at the bottom of the Early Triassic Daye Formation, Chibi West section, (**F**) Limestone at the top of the Late Permian Dalong Formation, Chibi West section.

## 3. Samples and Methods

The carbonate samples collected from the Chibi North and Chibi West profiles were first cut with a rock saw. Calcareous material, particularly calcite veins, was removed from the rock. The samples were cut into blocks and were polished on a vibrating gemstone tray before a 1 mm dental drill was used to select areas of the microcrystalline rock for drilling. Powder was drilled out for isotopic and elemental analysis. Powders of 78 samples were analyzed for carbon and oxygen isotopes, and of 29 samples for calcium isotopes, as well as for major and trace element analyses.

Calcium isotope analyses were performed at the State Key Laboratory of Geological Processes and Mineral Resources, China University of Geosciences, Wuhan. To obtain a reliable Ca isotope signature, a leaching procedure, modified from published methods [22], was designed to extract the pure carbonate fraction of the limestone samples. About 10 mg of the sample powder was weighed into 10 mL centrifuge tube and 1 mL MQ $H_2O$ and 3 mL 0.5 M $HNO_3$ were successively added. Following ca. 40 min vibration in ultrasonotor and standing for 12 h at room temperature to ensure complete reaction, the sample was centrifuged at 3500 rpm for 15 min. Subsequently, the supernatant liquid was moved to 7 mL Teflon vials and evaporated to dryness at 120 °C. Before chemical purification, dried samples were re-dissolved in 1.6 mL of 4 mol $L^{-1}$ $HNO_3$.

We followed the published procedures for chemical purification and mass spectrometric analyses [35] and only provide a brief description here. An aliquot sample solution of 40 ug Ca was loaded into the pre-cleaned column filled with 250 μL DGA resin (Ca-selective resin). Other elements present as matrix were removed by addition of 6.8 mL 4 mol $L^{-1}$ $HNO_3$ after loading the sample, and the Ca fraction was collected in 3 mL $H_2O$. We achieved a highrecovery (>99%), efficient separation of Ca, and a low total procedural blank of <10 ng. Calcium isotopes were analyzed using a Nu Plasma 1700 MC-ICP-MS [35]. Isotope measurements were performed using standard-sample bracketing to correct instrumental drift. The Ca isotopic compositions were reported by using δ-notation: $\delta^{x/42}Ca_{SRM915a} = [(^{x}Ca/^{42}Ca)sample/(^{x}Ca/^{42}Ca)_{SRM915a} - 1] \times 1000$, where, x = 44 or 43. Considering that $^{44}Ca/^{40}Ca$ is commonly used in literature, all Ca isotopic ratios in this study are reported as $\delta^{44/40}Ca$ where $\delta^{44/40}Ca = \delta^{44/42}Ca \times 2.048$.

The C-O isotope analysis of carbonate rocks was analyzed using the MAT253-kiel IV coupled online method in a fully automated process. Powder samples of microgram level were weighed in a reaction flask and connected to a vacuum system. After evacuation, 3–5 drops of high purity phosphoric acid were injected into the flask and the sample powder reacted with phosphoric acid at 70 °C while collecting $CO_2$ gas in trap1 with liquid nitrogen at −196 °C. Possible tramp gas in the vacuum system was removed, and subsequently trap1 was heated to −90 °C to release $CO_2$ and transferred to trap2 where a small amount of water was frozen in trap1. The trap2 was heated to 30 °C, and the released $CO_2$ diffused through the capillary tube to the MAT253 two-way system. The isotope ratio was determined by using a two-way injection method to analyze the standard VPDB. The results of repeated analysis showed that the accuracy of C isotope analysis of carbonate rocks was ±0.1‰, and the accuracy of O isotope analysis was ±0.2‰.

The major elements were measured using the XRF-18. The first, the samples were dried at 105 °C for 2 h in an oven. The dried samples were used to measure the loss on ignition (LOI) and to produce glass melts. Standard samples GBW07105 and GBW07407 were used for analysis. The Ca concentration was used to estimate the $CaCO_3$ content in the carbonates. The samples were analyzed at the State Key Laboratory of Biogeology and Environmental Geology, China University of Geosciences, Wuhan, P. R. China. Trace elements were analyzed by LA-MC-ICP-MS at the State Key Laboratory of Geological Processes and Mineral Resources, China University of Geosciences, Wuhan, P. R. China. Details on the analytical procedure are provided in [36].

## 4. Results

### 4.1. Carbon and Oxygen Isotopes

The carbon and oxgen isotope data are summarized in Table 1. The samples of the Chibi North and Chibi West profiles record the negative carbon isotope bias at the Permian-Triassic boundary (Figure 3). The analyses show that the $\delta^{13}C$ values of the Chibi North profile become negative at the top of the Dalong Formation, from 2.91‰ to −0.06‰ at the bottom of the Daye Formation, whereas they increase to 2‰ in the upper Daye Formation. The $\delta^{13}C$ valuesin the Chibi West proflile also becomes negative at the top of the Dalong Formation and decreases from 2.11‰. They reach −3.23‰ at the bottom of the Daye Formation and then increases again to about 0‰ in the upper part of the Daye Formation. Comparing the two profiles the $\delta^{13}C$ values of the Chibi North profile are generally higher than that of the Chibi west profiles.

**Table 1.** Element and isotope composition of the studied carbonate successions.

| Sample Number | Location | Strat Hight (cm) | $\delta^{13}C$ (PDB) | $\delta^{18}O$ (PDB) | $\delta^{44/40}$ Ca (SRM915a) | 2SD/‰ | n | $CaCO_3$ (%) | Mg/Ca | Mn/Sr | Sr/Ca ($\times 10^{-3}$) |
|---|---|---|---|---|---|---|---|---|---|---|---|
| Sam.1 | Chibi North | 1110 | 2.19 | −6.12 | 0.30 | 0.04 | 2 | 88.19 | 0.59 | 0.20 | 2.23 |
| Sam.2 | Chibi North | 1071 | 1.96 | −6.43 | | | | | | | |
| Sam.3 | Chibi North | 1037 | 1.85 | −6.57 | | | | | | | |
| Sam.4 | Chibi North | 993 | 1.87 | −6.26 | | | | | | | |
| Sam.5 | Chibi North | 950 | 1.75 | −6.27 | | | | | | | |
| Sam.6 | Chibi North | 855 | 1.56 | −5.79 | | | | | | | |
| Sam.7 | Chibi North | 820 | 1.88 | −5.78 | | | | | | | |
| Sam.8 | Chibi North | 789 | 1.82 | −5.95 | | | | | | | |
| Sam.9 | Chibi North | 750 | 1.58 | −5.91 | | | | | | | |
| Sam.10 | Chibi North | 675 | 1.51 | −6.15 | 0.72 | 0.13 | 3 | 89.69 | 0.03 | 0.34 | 1.92 |
| Sam.11 | Chibi North | 640 | 1.42 | −6.05 | | | | | | | |
| Sam.12 | Chibi North | 605 | 1.12 | −6.08 | | | | | | | |
| Sam.13 | Chibi North | 577 | 1.05 | −5.68 | | | | | | | |
| Sam.14 | Chibi North | 474 | 1.03 | −6.25 | | | | | | | |
| Sam.15 | Chibi North | 454 | 0.88 | −6.37 | | | | | | | |
| Sam.16 | Chibi North | 414 | 0.87 | −6.31 | 0.39 | 0.05 | 3 | 84.17 | 0.04 | 0.69 | 1.00 |
| Sam.17 | Chibi North | 374 | 0.72 | −6.30 | | | | | | | |
| Sam.18 | Chibi North | 345 | 0.69 | −6.33 | | | | | | | |
| Sam.19 | Chibi North | 314 | 0.51 | −6.40 | 0.48 | 0.02 | 3 | 84.58 | 0.04 | 0.68 | 1.01 |

**Table 1.** *Cont.*

| Sample Number | Location | Strat Hight (cm) | δ¹³C (PDB) | δ¹⁸O (PDB) | δ⁴⁴/⁴⁰ Ca (SRM915a) | 2SD/‰ | n | CaCO₃ (%) | Mg/Ca | Mn/Sr | Sr/Ca (×10⁻³) |
|---|---|---|---|---|---|---|---|---|---|---|---|
| Sam.20 | Chibi North | 270 | 0.38 | −6.31 | | | | | | | |
| Sam.21 | Chibi North | 241 | 0.36 | −6.69 | | | | | | | |
| Sam.22 | Chibi North | 209 | 0.29 | −6.18 | | | | | | | |
| Sam.23 | Chibi North | 176 | 0.18 | −6.15 | | | | | | | |
| Sam.24 | Chibi North | 155 | 0.29 | −6.51 | | | | | | | |
| Sam.25 | Chibi North | 130 | −0.06 | −6.28 | 0.77 | 0.08 | 3 | 65.46 | 0.09 | 1.81 | 0.98 |
| Sam.26 | Chibi North | 114 | 0.35 | −5.57 | | | | | | | |
| Sam.27 | Chibi North | 100 | 0.44 | −6.20 | | | | | | | |
| Sam.28 | Chibi North | 69 | 0.27 | −5.01 | 0.76 | 0.13 | 2 | 20.07 | 0.65 | 3.70 | 1.56 |
| Sam.29 | Chibi North | 55 | 0.65 | −4.06 | | | | | | | |
| Sam.30 | Chibi North | 44 | 0.78 | −6.61 | | | | | | | |
| Sam.31 | Chibi North | 35 | 1.41 | −4.25 | 0.48 | 0.48 | 2 | 56.28 | 0.49 | 3.86 | 0.89 |
| Sam.32 | Chibi North | 27 | 2.37 | −1.37 | 0.78 | 0.14 | 5 | 55.40 | 0.54 | 4.63 | 0.91 |
| Sam.33 | Chibi North | −5 | 2.50 | −5.56 | 0.51 | 0.08 | 3 | 54.08 | 0.64 | 5.83 | 0.68 |
| Sam.34 | Chibi North | −65 | 2.91 | 0.25 | 0.91 | 0.13 | 3 | 54.98 | 0.59 | 5.85 | 0.66 |
| Sam.35 | Chibi North | −145 | 1.66 | −7.62 | 0.78 | 0.21 | 3 | 66.46 | 0.10 | 2.26 | 0.77 |
| Sam.36 | Chibi North | −215 | 2.00 | −8.47 | | | | | | | |
| Sam.37 | Chibi North | −290 | 2.44 | −7.51 | 0.82 | 0.07 | 3 | 65.69 | 0.05 | 1.13 | 0.78 |
| Sam.38 | Chibi North | −435 | 2.51 | −7.85 | | | | | | | |
| Sam.39 | Chibi North | −519 | 2.71 | −8.75 | 0.60 | 0.04 | 3 | 43.74 | 0.03 | 0.68 | 0.65 |
| Sam.40 | Chibi North | −595 | 2.28 | −5.69 | | | | | | | |
| Sam.41 | Chibi North | −675 | 2.32 | −10.30 | 0.42 | 0.17 | 3 | 23.84 | 0.03 | 0.68 | 1.19 |
| Sam.42 | Chibi North | −745 | 1.86 | −10.09 | | | | | | | |
| Sam.43 | Chibi North | −895 | 1.86 | −8.77 | | | | | | | |
| Sam.44 | Chibi North | −1130 | 2.14 | −8.58 | | | | | | | |
| Sam.45 | Chibi North | −1220 | 2.99 | −5.85 | 0.56 | 0.17 | 3 | 75.28 | 0.01 | 1.52 | 0.34 |
| Sam.46 | Chibi North | −1270 | 3.03 | −8.92 | | | | | | | |
| Sam.46 | Chibi West | 1228 | 0.09 | −7.86 | 0.41 | 0.08 | 3 | 96.22 | 0.018 | 0.11 | 1.84 |
| Sam.47 | Chibi West | 1168 | 0.04 | −7.10 | | | | | | | |
| Sam.48 | Chibi West | 1098 | −0.23 | −9.53 | | | | | | | |
| Sam.49 | Chibi West | 1018 | 0.28 | −7.82 | | | | | | | |
| Sam.50 | Chibi West | 938 | −0.16 | −7.69 | 0.29 | 0.05 | 3 | 95.52 | 0.016 | 0.08 | 2.49 |
| Sam.51 | Chibi West | 858 | −0.02 | −7.98 | | | | | | | |
| Sam.52 | Chibi West | 778 | −0.35 | −8.63 | 0.38 | 0.03 | 2 | 91.10 | 0.021 | 0.43 | 1.00 |
| Sam.53 | Chibi West | 698 | −0.27 | −7.81 | | | | | | | |
| Sam.54 | Chibi West | 618 | −0.13 | −8.08 | 0.27 | 0.07 | 2 | 92.81 | 0.019 | 0.28 | 1.48 |
| Sam.55 | Chibi West | 538 | −1.33 | −8.24 | | | | | | | |
| Sam.56 | Chibi West | 483 | −1.74 | −9.02 | 0.29 | 0.05 | 2 | | 0.050 | 1.51 | 1.11 |
| Sam.57 | Chibi West | 403 | −1.73 | −9.81 | | | | | | | |
| Sam.58 | Chibi West | 353 | −3.23 | −8.93 | 0.39 | 0.09 | 3 | 11.55 | 0.561 | 3.72 | 1.80 |
| Sam.59 | Chibi West | 303 | −2.74 | −9.83 | | | | | | | |
| Sam.60 | Chibi West | 253 | −1.91 | −7.10 | | | | | | | |
| Sam.61 | Chibi West | 203 | −1.88 | −6.84 | 0.32 | 0.07 | 3 | 27.28 | 0.534 | 5.72 | 1.12 |
| Sam.62 | Chibi West | 153 | −1.31 | −8.04 | | | | | | | |
| Sam.63 | Chibi West | 103 | −1.74 | −10.24 | | | | | | | |
| Sam.64 | Chibi West | 88 | −0.93 | −6.91 | 0.43 | 0.07 | 3 | 47.80 | 0.402 | 2.89 | 0.98 |
| Sam.65 | Chibi West | 58 | −1.07 | −9.61 | | | | | | | |
| Sam.66 | Chibi West | 28 | −0.09 | −8.71 | | | | | | | |
| Sam.67 | Chibi West | 8 | 0.57 | −9.25 | 0.54 | 0.08 | 3 | 88.22 | 0.021 | 0.56 | 1.18 |
| Sam.68 | Chibi West | −15 | 0.99 | −10.34 | 0.60 | 0.11 | 3 | 98.06 | 0.017 | 0.29 | 1.37 |
| Sam.69 | Chibi West | −30 | 2.11 | −9.47 | | | | | | | |
| Sam.70 | Chibi West | −53 | 2.22 | −9.57 | 0.36 | 0.01 | 3 | 96.95 | 0.020 | 0.17 | 1.17 |
| Sam.71 | Chibi West | −83 | 2.32 | −9.64 | | | | | | | |
| Sam.72 | Chibi West | −113 | 2.54 | −8.84 | 0.43 | 0.09 | 3 | 97.18 | 0.020 | 0.15 | 1.34 |
| Sam.73 | Chibi West | −130 | 2.46 | −9.42 | | | | | | | |
| Sam.74 | Chibi West | −158 | 2.79 | −10.37 | | | | | | | |
| Sam.75 | Chibi West | −188 | 2.82 | −9.68 | 0.60 | 0.06 | 3 | 94.88 | 0.019 | 0.14 | 1.42 |
| Sam.76 | Chibi West | −218 | 2.33 | −7.78 | | | | | | | |
| Sam.77 | Chibi West | −248 | 2.28 | −8.88 | 0.18 | 0.10 | 3 | 92.83 | 0.021 | 0.13 | 1.55 |
| Sam.78 | Chibi West | −278 | 2.40 | −8.63 | | | | | | | |

The oxygen isotope valuesof Chibi North range from −10.3 to −0.25‰. The oxygen isotope values of Dalong Formation rapidly increase from −7.5‰ at the MEB (mass extinction boundary) to −0.25‰ near the PTB (Permian-triassic boundary), subsequently decrease to −6‰ after passing the Permian-triassic boundary and remain stable in the

Triassic. The oxygen isotope values of Chibi West profile generally resemble the course of the isotope values of the Chibi North profile. They increase to −6.5 between the MEB and PTB and decrease back to −9.5‰ before MEB.

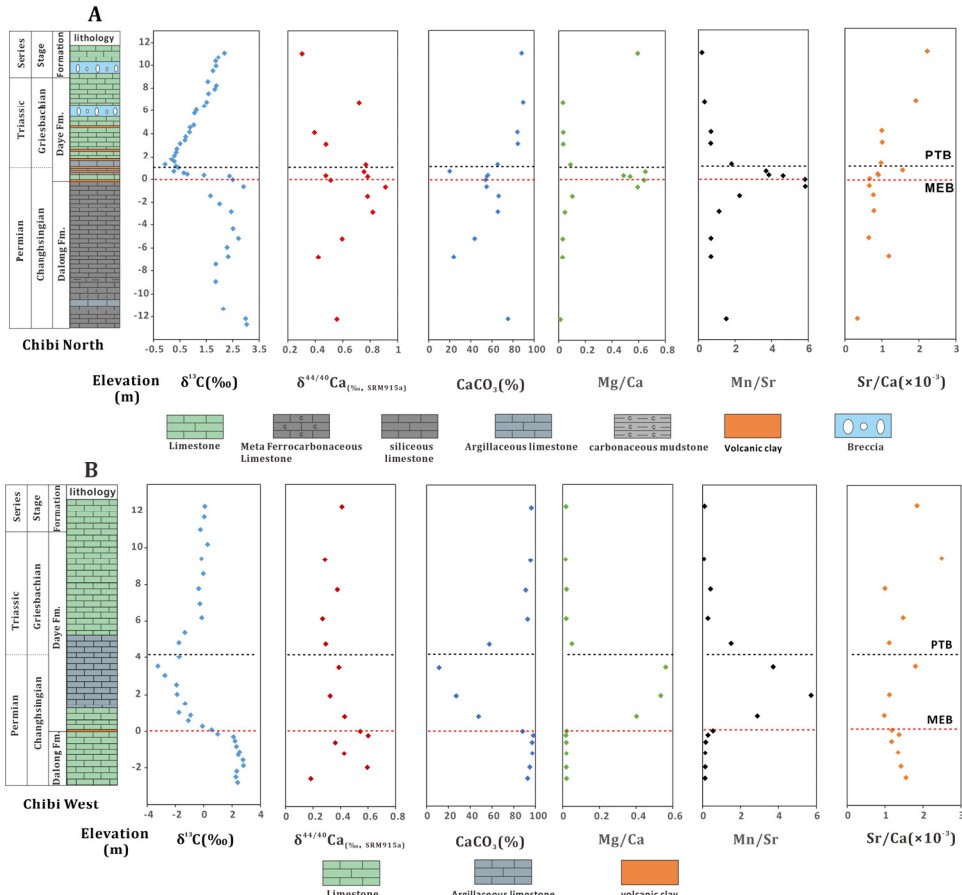

**Figure 3.** (**A**) Lithostratigraphy and chemostratigraphy of the Chibi North section, (**B**) Lithostratigraphy and chemostratigraphy of the Chibi West section (MEB—mass extinction boundary, PTB—Permian-Triassic boundary).

*4.2. Calcium Isotopes*

The calcium isotope data are also presented in Table 1. The calcium isotope values of the Chibi North and Chibi West profiles show a certain degree of negative bias from the mass extinction boundary to the Permian-Triassic boundary (Figure 3 & Table 1). The $\delta^{44/40}$Ca value of the Chibi North profile ranges from 0.41‰ to 0.91‰ in the Late Permian. The δ44/40Ca value of Chibi North decreases by 0.47‰ from 0.91‰ before MEB, and increases again to 0.76% after the PTB. Subsequently, it varies between 0.3‰ and 0.76‰ in the Griesbachian stage. The $\delta^{44/40}$Ca value of Chibi West in the Late Permian ranges from 0.18‰ to 0.60‰, and after the MEB decreases from 0.60‰ to 0.32‰. In the Early Triassic the δ44/40Ca value varies between 0.27‰ and 0.41‰.

*4.3. Major and Trace Elements*

Results of the major element analyzes for the rocks of the Chibi North and Chibi West profiles are illustrated for the $CaCO_3$ content as well as the Sr/Ca ($\times 10^{-3}$), Mg/Ca, and Mn/Sr ratios in Figure 3 and summarized in Table 1. The $CaCO_3$ content decreases dramatically in the samples after the MEB in both, the Chibi North and Chibi West profiles. In the Chibi North profile, it decreases from 55% to 20%, and subsequently increases to 80% in the subsequent Triassic. In the Chibi West profile, it drops from 98% to 11% and rises to 92% in the Triassic. The Mg/Ca, Mn/Sr, and Sr/Ca ($\times 10^{-3}$) ratios of the Chibi North and Chibi West sections all increase significantly after the extinction line (Figure 2). The Mg/Ca

ratios sections are less than 0.1 prior to the MEB in both, Chibi North and Chibi West, but increase to 0.6 after the extinction line., in the subsequent Triassic, the value decreases again to <0.1. The Mn/Sr ratio in Chibi North increases substantially from 3 to 8, during the MEB and decreases to <2 in the Triassic. The Mn/Sr ratios of Chibi West increases from 0.5 to 5.7 when passing through the MEB, and decline to 0.2 in the Triassic. The Sr/Ca ($\times 10^{-3}$) in Chibi North increases from 0.6 to 1.5 after passing through the MEB and decrease to 1 in the Triassic. The value of the Sr/Ca ($\times 10^{-3}$) ratio in Chibi West increase from 1.1 to 1.8 after the MEB, and fluctuate between 1.1 and 2.4 in the Triassic. The specific data is summarized in Table 1.

## 5. Discussion

### 5.1. Impact of Diagenesis on $\delta^{44/40}Ca$ Values

During the diagenesis of carbonate rocks, the initial metastable carbonate minerals attain a stable state through structural, compositional, and chemical changes [37]. Specific elements ($Na^+$, $Sr^{2+}$, $Mn^{2+}$, $Mg^{2+}$, and $Fe^{2+}$) can substitute $Ca^{2+}$ in carbonate minerals, such as calcite and aragonite, by several processes, including metasomatism, pore filling, adsorption, and filling of crystal defects, The substitution causes change of the chemical compositions of the carbonate rocks [38]. Aragonite can be converted into calcite. The position of ions in calcite is much smaller than that in aragonite. Therefore, the small $Mg^{2+}$ can substitute $Ca^{2+}$ in calcite, whereas larger the $Sr^{2+}$ can easily substitute $Ca^{2+}$ in aragonite [39]. The relationship between element ratios (Mg/Ca and Sr/Ca) and the calcium isotopes ($\delta^{44/40}Ca$) can be used as a tool for identifying the diagenetic transformation of carbonate rocks [40]. In marine environments, seawater is usually the highest end number with $\delta^{44/40}Ca$ value [13]; Various original carbonate minerals are enriched in light calcium isotopes when precipitated from seawater, then have lower calcium isotope values [13,19]. Among them, native aragonite has the characteristics of lower calcium isotope and higher Sr/Ca value [41,42]; Calcite is characterized by relatively high calcium isotopes and low Sr/Ca values [41,42]; dolomite has the characteristics of higher calcium isotopes and lower Sr/Ca values [43,44]. Based on the correlation between carbonate $\delta^{44/40}Ca$ values and Sr/Ca, diagenesis can be qualitatively assessed [36] The early diagenesis of carbonate rocks of the same geological age shows quantitative covariation between the value of $\delta^{44/40}Ca$ and the ratio of Sr/Ca in the correlation diagram. The ratio of Sr/Ca decreases with rising $\delta^{44/40}Ca$ values [21]. The Ca isotope composition in carbonate rocks changes due to dolomitization. With rise of the Mg/Ca ratio the $\delta^{44/40}Ca$ value increases [5]. The Mn/Sr ratio of carbonate rocks is often used to constrain the degree of diagenetic transformation of carbonate rocks [45] and water-rock interaction will reduce the $\delta^{18}O$ value of carbonate rocks [46].

Five calcium isotope profiles for carbonate rocks had been presented for the Permian-Triassic boundary in previous studies [5,14,22]. Two sections are from southern China, including a section for the Dajiang section in the Guizhou Basin, which is a shallow-water carbonate platform section [14]. The second section was obtained for the MeiShan section in the Zhejiang Province, which is situated in the carbonate platform slope facies region of the Lower Yangtze region [22]. The Taskent section in Turkey is situated inside a shallow-water carbonate platform [47]. Silva-Tamayo [5] analyzed calcium isotopes of the carbonate rocks in the Saiq and Tesero sections. The Saiq section is located in Al Jabal Al-Akhdar (Sultanate of Oman) and represents a carbonate platform dominated by dolomite. The rocks of the Tesero profile (Italy) were deposited at shallow depths on a carbonate ramp. We compare the Chibi West and Chibi North profiles with these five previously published profiles in Figure 3. The calcium isotope value of the Chibi North section is 0.23‰ higher than that of the Chibi West section on average. The Ca/Sr ratio of the Chibi West section is lower than that of the Chibi West section (Figure 4), indicating that the Chibi West section was affected by more severe early diagenesis. The ratio of Mg/Ca in the Chibi North section is higher than that in the Chibi West section, documenting more intense dolomitization of the rocks from the Chibi North section. The Saiq section is a dolomite-dominated carbonate rock

section that records the highest Mg/Ca ratios and $\delta^{44/40}$Ca values, indicating the rocks were affected by the most intense dolomitization of all sections. The rocks of the Chibi North profile have higher $\delta^{18}$O values than that of the western profile (Figures 3 and 4). Studies indicate that $\delta^{18}$O values < −5‰, and especially values < −10‰, indicate that the marine carbonate deposits have been affected by diagenetic fluids from atmospheric precipitation. In contrast, $\delta^{18}$O values > −5‰ document that carbonate deposits were not or only moderately modified by atmospheric precipitation [48]. The data indicate that the rocks of the Chibi West section were affected by atmospheric precipitation, whereas the impact of atmospheric precipitation is negligible for the rocks Chibi North section.

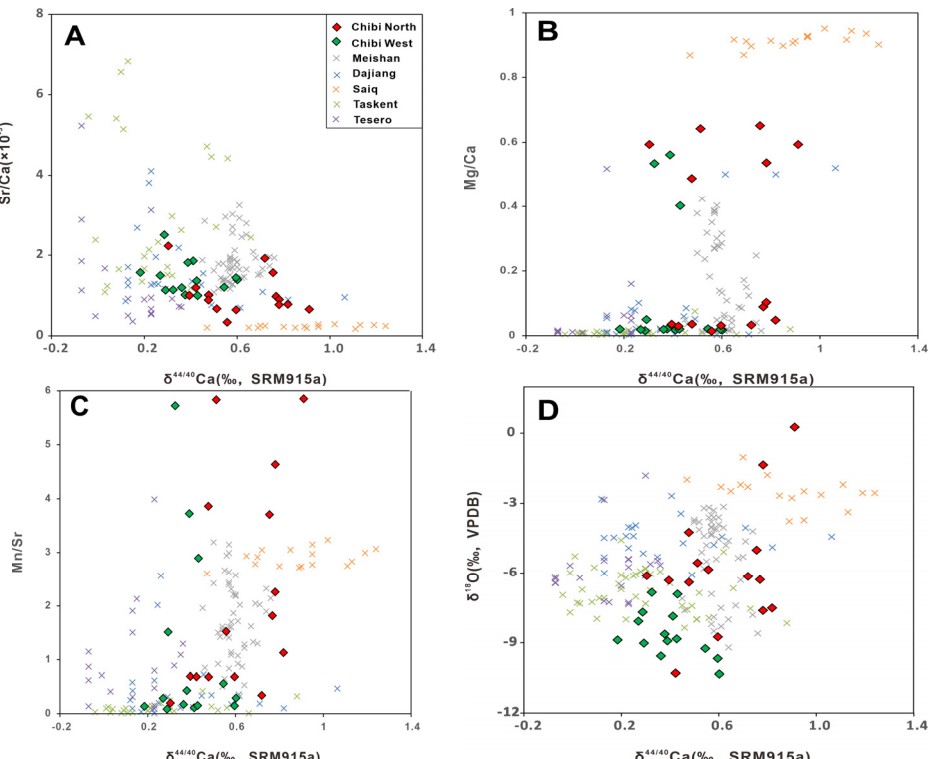

**Figure 4.** Scatter plots(**A**) Ca/Sr vs. $\delta^{44/40}$Ca; (**B**) Mg/Ca vs. $\delta^{44/40}$Ca; (**C**) Mn/Sr vs. $\delta^{44/40}$Ca; (**D**) $\delta^{18}$O vs. $\delta^{44/40}$Ca.Reflecting the impact of the diagenesis. Filled colored squares indicate data from this study. Colored forks indicate data from other profiles (Saiq, Tesero, Taskent data are from [5], Dajiang section are data from [14], and Meishan section data are from [22]) The data conversions for calcium isotopes are based on ($\delta^{44/40}$Ca$_{915a}$ = $\delta^{44/40}$Ca$_{sw}$ + 1.86‰, $\delta^{44/40}$Ca$_{BSE}$ = $\delta^{44/40}$Ca$_{915a}$ −1.00‰) [49].

### 5.2. The C-Ca Cycle at Chibi

The coupling of the carbon cycle with the calcium cycle is mainly related to the burial of calcium carbonate and the alteration of calcium-bearing minerals. The $\delta^{44/40}$Ca and $\delta^{13}$C records from the same period can be used to estimate the amount of carbon required for the negative excursion of $\delta^{44/40}$Ca and $\delta^{13}$C to verify the conjecture of ocean acidification [50]. The negative carbon and calcium isotope biases between MEB and PTB of the Chibi North and Chibi West profiles correspond to the global negative C-Ca isotopic data during this period. Based on our results of diagenesis-related indicators (Sr/Ca, Mg/Sr, Mn/Sr, $\delta^{18}$O), we propose that the Chibi North and Chibi West sections were transformed by diagenesis to varying degrees. In addition to diagenesis, perturbations in the marine carbonate chemistry can also affect the $\delta^{44/40}$Ca and $\delta^{13}$C values. According to a coupled box model of geological carbon and calcium cycles [51,52], carbonate burial rates increase with increasing alkalinity. This leads to a rise in the seawater carbonate saturation, inducing that the values of $\delta^{44/40}$Ca in the seawater and $\delta^{44/40}$Ca in the carbonate sediments are both

positively skewed. Conversely, ocean acidification due to increased $CO_2$ release related to volcanism would dissolve calcium carbonate, leading to a negative bias in the $\delta^{44/40}Ca$ value in the carbonate rocks. In the coupled model with negatively biased $\delta^{44/40}Ca$ and $\delta^{13}C$, the value of $\delta^{44/40}Ca$ can only be negatively biased by 0.13‰ at most, irrespective of the amount and duration of $CO_2$ input [5]. Therefore, the ocean acidification caused by $CO_2$ released by volcanic activity cannot explain the negative deviations of $\delta^{44/40}Ca$ values. When the proportion of aragonite in the two Chibi sections is high, the Sr/Ca ratio will also rise to a higher value between MEB and PTB at this time. A major decrease of the $CaCO_3$ amount was observed between MEB and PTB in Chibi North and Chibi West. The decrease is probably related to the large-scale release of $CO_2$ caused by the widespread volcanic activity at the Permian-Triassic boundary. Volcanism induced ocean acidification that in turn caused intense dissolution of calcium carbonate and the negative bias of calcium isotopes. However, additional processes are required to explain the calcium isotope characteristics that we observed in the Chibi North and Chibi West profiles.

*5.3. Calcium Isotope Signatures at Different Water Depths*

Significant differences of the $\delta^{44/40}Ca$ values are observed for the five published profiles and our two Chibi section at the Permian-Triassic boundary (Figure 5). Comparison of the calcium isotope values at different water depths in the Chibi North and Chibi West sections indicates an increase of the value of the calcium isotope of the carbonate rock with increasing water depth (Figure 5). The residence time of calcium in the ocean is about $(0.5{\sim}1 \times 10^6)$ years [53], which is longer than the mixing time of seawater (about $1{\sim}2 \times 10^3$ years) [54]. Therefore, modern seawater shows generally a uniform $\delta^{44/40}Ca$ value of ca. 1.88‰ [55]. The calcium isotope values at different water depths are characterized by gradients (Figure 5). The increase of the calcium isotope value with increasing water depth on slopes is therefore apparently related to different depositional environments.

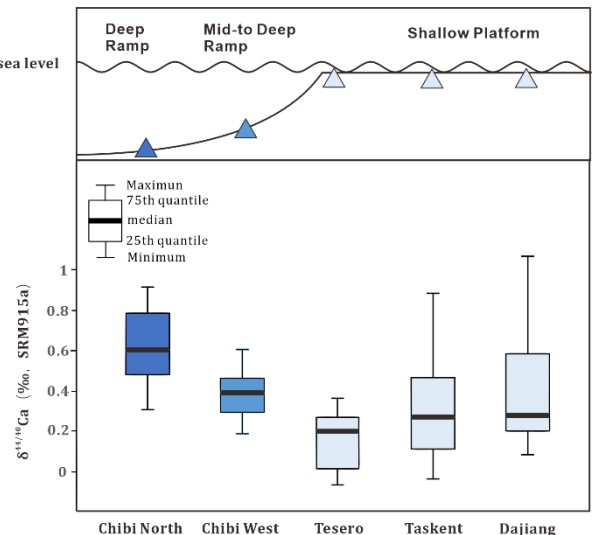

**Figure 5.** Calcium isotope data at different water depths during the Permian-Triassic. The colorful triangles represent the carbonate section. Chibi North and Chibi West sections data are from this study. Tesero, Taskent data are from [5], Dajiang section data are from [14].

Since the value of $\delta^{44/40}Ca$ is affected by diagenesis, the ratio of aragoniteand calcite in the sediment will specifically influence the value of $\delta^{44/40}Ca$. The calcium isotope value of Chibi North is higher than that of Chibi West on average, and Chibi North has a lower Ca/Sr ratio. The data document that the carbonate rocks in Chibi North suffered more severe early diagenesis. The velocities of interstitial fluids are typically the highest in the shallow water of the slopes, whereas the deep water is comparably calm [56]. Intense

interstitial fluids in shallow water promote diagenesis. We propose that this cross-regional calcium isotope gradient is related to the impact of pore fluids on the diagenesis that induced the transformation of aragonite deposited in the primary carbonate rocks into calcite by early diagenesis. We propose the following scenario to explain the observed calcium isotope gradients.

Prior to the fall of the sea level, the present-day Chibi North and Chibi West profiles were situated in deep water and aragonite was the product of primary carbonate precipitation. With the drop of the sea level, the Chibi West profile was exposed to the surface and aragonite in the carbonate rocks was preserved. Due to the retreat of the coastline, the sedimentary environment of the Chibi North section changed from deep water to shallow water, and through pervasive ingress of pore fluid aragonite was transformed into calcite, inducing the increase of $\delta^{44/40}$Ca value(Figure 6B).The aragonite in the carbonate rocks in the shallow sections of Italy and Turkey was the first to avoid the erosion of the nearshore pore water due to the subsidence process, with better primary aragonite preservation, also the erosion of the pore fluid in the whole process was less than that in the of Chibi West section(Figure 6B). These sections were subjected to different diagenesis when sea level goes down and the carbonate rocks deposited under them were preserved. The less diagenesis at the approximate shallow water level, and the diagenetic information at that time is better preserved (Figure 6C). The value of $\delta^{44/40}$Ca was recognized to decrease with the increase of the water depth in carbonate rocks at different water depths in rocks from the Upper Ordovician of Nevada and Utah (USA) basin, when the basin was subjected to transgression [21]. This is consistent with our results of the Chibi slope and we propose a comparable tectonic scenario.

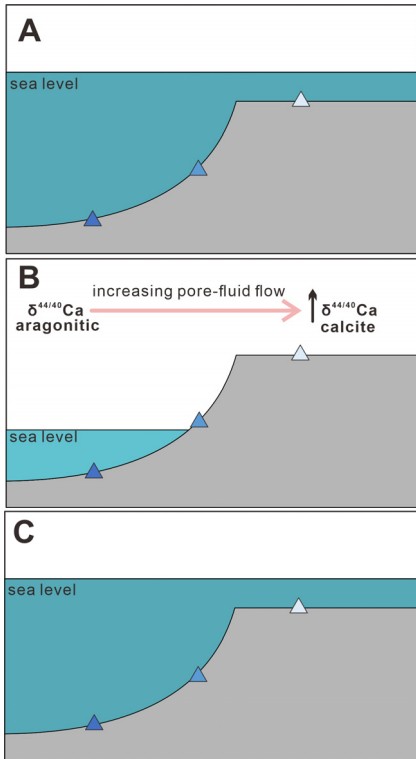

**Figure 6.** Simulated geochemical changes during sea level fall in the Chibi region (**A**) Before the Late Permian sea level decline, (**B**) Late Permian sea level decline and (**C**) Early Triassic sea level rebound. The colorful triangles represent the carbonate section.

## 6. Conclusions

The calcium isotope values of the Chibi North and Chibi West sections were negatively skewed after the Permian-Triassic mass extinction line, consistent with the global negative

calcium isotope skew during this period. The Chibi North section has higher calcium isotope values than the Chibi West section, whereas the Sr/Ca ratio of the Chibi North section is lower. These data indicate that the carbonate rocks in the Chibi West suffered less intense early diagenesis, causing the better preservation of primary aragonite. Based on the comparison of diagenesis indicators (Sr/Ca, Mg/Ca, Mn/Sr, $\delta^{18}$O), we propose that both, the Chibi West and Chibi North sections were affected by diagenesis. The negative bias of calcium isotopes in the Chibi area cannot be solely explained by ocean acidification but are stronger influenced by diagenesis. Through the integration of the three previously published shallow water profiles of carbonate rocks for this period, we recognized a gradient of calcium isotopes that is related to the enhancement of pore fluids during early diagenesis, which caused the transformation of primary aragonite to calcite in the carbonate rocks. The calcium isotopes of salt rocks at different water depths can reflect fluid buffer alteration, especially the enhancement of interstitial fluid near the shore of the slope caused by sea level changes in the marine slope environment. We propose that a regression event occurred in the Chibi region at the end of the Permian. Our case study for the Chibi region indicates that calcium isotopes in carbonate rocks have the potential to record the rise and drop of the sea level.

**Author Contributions:** Conceptualization: H.Z. and J.H.; methodology: H.Z.; validation: H.Z. and J.H.; investigation: H.Z.; data curation: H.Z.; writing—original draft preparation: H.Z.; writing—review and editing: H.Z. and H.Z. All authors have read and agreed to the published version of the manuscript.

**Funding:** This research was funded by National Natural Science Founding of China (Grant No. 41730320).

**Data Availability Statement:** Not applicable.

**Conflicts of Interest:** The authors declare no conflict of interest.

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
