# Peer review of "Characteristics of Calcium Isotopes at Different Water Depths and Their Palaeoenvironmental Significance for Carbonate Rocks of the Permian-Triassic Boundary in Chibi, Southern China"

_minerals, doi:10.3390/min12111440_

Round 1
Reviewer 1 Report
This manuscript presents a detailed investigation of calcium, carbon and oxygen Isotopes at different water depths in Chibi, Southern China, to understand environment during the Permian-Triassic mass extinction period. The paper is well prepared and easy to read. I recommend that this paper should be accepted after minor revisions. The comments are listed below for authors' consideration.
1. Please attach profile map, pictures of outcrop and hand samples.
2. Figure 3 should be rearranged and making them more orderliness.
3. Figure 4, the bottom line of the box in the legend of the box plots should be “25th quantile”, not “15th”.
4. Figure 5, to add a scale bar. There should be quantitative data for sea level rise and fall.
Author Response
-
Dear reviewer, attached is the reply letter.

Reviewer 2 Report
Dear authors,
herewith my comments on your manuscript entitled "Characteristics of Calcium Isotopes at Different Water Depths and Their Palaeoenvironmental Significance for Carbonate Rocks of the Permian-Triassic Boundary in Chibi, Southern China". The aim of the study is to analyse the C, O, Ca isotopes and major and trace elements of two carbonate successions spanning the Permian-Triassic boundary in order to provide new data for a better understanding of the global events characterizing this time interval.
The manuscript is based on an adequate dataset and the quality of the images provided is satisfactory, but not a single picture of the studied sections has been provided. Furthermore, the reference list is extremely limited and the inferred interpretations are not adequately supported nor discussed. Last but not least, the text needs to be heavily reorganized and rewritten, and at least partly rethought. The manuscript is very difficult to read and understand as it is. I suggest rejecting the manuscript.

Author Response
Dear reviewer, attached is the reply letter

Reviewer 3 Report
The manuscript has been developed with poor English level and need to be essentially improved.
1. Line-13: Please correct the word “environment”.
2. Line-22: Please explain more clearly how diagenesis affects isotopes.
3. Line-56: Please correct the word “circulation”.
4. Figure 1 seems to be stretched. Please keep the original scale. And the legend also needs to be as normalized as possible.
5. Where are Chongyang, Hubei and Tongshan in Figure 1? Please mark their positions in Figure1. Besides, the location of the two sections should also be marked.
6. Line-98 and Line-129: There are very serous grammatical errors.
7. The stratigraphic column and tectonic history of the study area need to be supplemented.
8. Line-152: “main elements” should be “major elements”.
9. Line-171: A comma is missing. There are too many grammatical errors in the whole article. Please check it carefully.
10. What is the relationship between PTB and major with trace elements in Section 4.3?
11. What does the “SD” mean in Table 1?
12. You should discuss the influencing factors of δ44/40Ca in the Section 5.1 not in the Introduction. The specific reasons for covariation between element ratios and calcium isotopes should also be given.
13. From Line 289 to Line 290, poor English expression makes it impossible to understand variations of geochemical parameters.
14. Line 309: What is the “agaric”? A poor spelling mistake.
15. What is the difference in aragonite/calcite ratio between those sections? You should discuss it and analyze its influence on calcium isotope.
16. Figure 5B and C have not been mentioned in this article. In addition, the explanation of calcium isotope changes at different depths is too simple to draw conclusion.
Author Response

(The authors gave the same response as above.)

Round 2
Reviewer 3 Report
After carefully modification, the manuscript is significantly improved, and suitable for publication.